

# Expression and prognostic analyses of early growth response proteins (EGRs) in human breast carcinoma based on database analysis

Yuchang Fei[1], Huan Yu[2], Shuo Huang[3], Peifeng Chen[1] and Lei Pan[1]

[1] The First Affiliated Hospital of Zhejiang Chinese Medical University, Hangzhou, Zhejiang Province, China
[2] Ningbo Yinzhou Second Hospital, Ningbo, Zhejiang Province, China
[3] The Third Clinical Medical Institute of Zhejiang Chinese Medical University, Zhejiang Province, China

## ABSTRACT

**Background**. Early growth response proteins (EGRs), as a transcriptional regulatory family, are involved in the process of cell growth, differentiation, apoptosis, and even carcinogenesis. However, the role of EGRs in tumors, their expression levels, and their prognostic value remain unclear.

**Methods**. Using the Oncomine database, Kaplan–Meier Plotter, bcGenExMiner v4.2, cBioPortal, and other tools, the association between the survival data of breast carcinoma (BC) patients and transcriptional levels of four EGRs was investigated.

**Results**. According to the Oncomine database, in comparison to normal tissues, the expression level of EGR2/3 mRNA in BC tissues was decreased, but there was no difference in the expression level of EGR4 mRNA. On the basis of the Scarff-Bloom-Richardson (SBR) grading system, the downregulated expression level of EGR1/2/3 and upregulated expression level of EGR4 were correlated with an increased histological differentiation level, with significant differences ($p < 0.05$). Kaplan–Meier curves suggest that a reduction in EGR2/3 mRNA expression is related to recurrence-free survival (RFS) in BC patients. In addition, the mRNA expression level of EGR1/2/3 was related to metastatic relapse-free survival (MRFS) in BC patients with metastatic recurrence ($p < 0.05$).

**Conclusion**. EGR1/2/3 can be utilized as an important factor for evaluating prognosis and may be relevant to diagnosis. EGR4 may play a role in the occurrence and development of BC. The specific function and mechanism of EGRs in BC deserve further study.

Corresponding author
Lei Pan, panlei21316@163.com

## INTRODUCTION

Breast carcinoma (BC) is currently the major cause of cancer-related deaths in women worldwide. According to 2012 data, there are approximately 1.7 million new cases and over half a million deaths each year. BC alone accounts for 25% of all cancer cases and 15% of cancer deaths in women. Worldwide, the incidence and mortality related to BC ranks first among female cancers (*Torre et al., 2015*). In many Asian and African countries,

morbidity and mortality have been increasing compared to those in Europe and the United States. This finding may be ascribed to westernized lifestyle changes and a lack of BC screening activities (*DeSantis et al., 2014*). BC has a variety of clinical, pathological, and molecular characteristics. The detection of the biological markers of BC, such as progesterone receptor (PR), estrogen receptor (ER), and tyrosine kinase ErbB2 receptor (HER2), can better predict the treatment and prognosis of patients (*Onitilo et al., 2009*).

Early growth response proteins (EGRs) are a transcriptional regulatory family and include EGR1/2/3/4. These proteins possess the ability to bind GC-rich recognition motifs in DNA (*Gashler & Sukhatme, 1995*) and mediate the processes involved in cell growth, differentiation, and apoptosis. As an anticancer gene, EGR1 has been observed and verified in numerous cancers. It has also been completely absent in breast cancer and lung cancer (*Huang et al., 1995*; *Ronski et al., 2010*). EGR1 plays a biological role in tumor cells by regulating the transcription of the heparin enzyme and either an inhibitory or an activation role in different tumor types (*de Mestre et al., 2005*). *Chen et al. (2017)* successfully inhibited the proliferation of glioma by knocking out EGR1. EGR1 is highly expressed in prostate cancer and is deemed to be the key factor that drives tumor progression (*Ma et al., 2009*). In ER+ breast tumors that were treated with endocrine therapy, the higher the expression level of EGR1 was, the better the prognosis (*Shajahan-Haq et al., 2017*). In addition, EGR2 has been discovered to inhibit the growth and invasion of SGC-7901 cells in gastric cancer, which implies that it may have an anticancer effect (*Chen et al., 2016*). Specifically, the researchers (*Chen et al., 2019*) inhibited gastric cancer metastasis and the epithelial-mesenchymal process by upregulating the expression of EGR2. EGR3 has been recognized to play a vital role in the invasion of BC and is an independent prognostic factor for BC (*Suzuki et al., 2007*). The upregulation of EGR3 improves the survival and proliferation ability of hepatocellular carcinoma (HCC) cells and promotes the migration and invasion of HCC cells (*Baron et al., 2015*). *He et al. (2019)* confirmed that the upregulation of EGR4 may promote the growth of non-small cell lung cancer (NSCLC) through the positive feedback regulatory circuit formed between ZNF205-AS1 and EGR4. Based on previous studies, EGR2, EGR3, and EGR4 were described as key regulators of T-cell activation in vivo and in vitro (*Williams et al., 2017*). Their potential role in cancer has been receiving increasing attention.

With the development of genomics, the genetic map of BC has continued to gain improvement. However, the identification of effective gene therapy targets for BC has become an urgent matter. This study examined the expression of the EGR gene family in tumor databases. Additionally, their expression in BC was analyzed to elucidate its value in the treatment and diagnosis of BC.

## MATERIALS & METHODS

### Oncomine database analysis

Oncomine (https://www.oncomine.org) is a large tumor gene microarray database covering 65 gene chip datasets, 4,700 chips and 480 million gene expression data points that can be used to analyze gene expression differences (*Rhodes et al., 2004*). The specific retrieval

parameters used were as follows: Retrieve: EGR1/2/3/4; Data type: mRNA; Analysis type: cancer and normal analysis; $P$ value: 0.0001; Fold change: 2.0; Gene rank: 10%; Analysis execution time: 2019.06.29. These data were collected from significantly different studies.

## The Cancer Genome Atlas (TCGA) and the Molecular Taxonomy of Breast Cancer International Consortium (METABRIC)

The Cancer Genome Atlas (TCGA) (https://portal.gdc.cancer.gov/) is a cancer gene database that enables genome sequencing and bioinformatics analyses through high-throughput genome analysis technology and includes 39 different cancer types. The Molecular Taxonomy of Breast Cancer International Consortium (METABRIC) (http://molonc.bccrc.ca/aparicio-lab/research/metabric/) is a project established by Canada and the UK that aims to classify BC according to its molecular characteristics to obtain the best clinical treatment. We used cBioPortal to visualize the data from the TCGA and METABRIC databases.

## cBioPortal

cBioPortal (https://www.cbioportal.org/) is a visualized analysis tool set that contains the genetic expression data and pathological information of 3617 BC patients. It further analyzes the expression of EGR family members through the exploration of multidimensional cancer gene set data. The retrieval parameters used were as follows: Analysis of cancer: breast cancer; Data set: Breast Cancer (METABRIC, Nature 2012 & Nat Commun 2016), Breast Invasive Carcinoma (TCGA, Provisional); Select Genomic Profiles: Mutations and copy-number alterations; Enter Genes: EGR1, EGR2, EGR3, EGR4.

## Kaplan–Meier plotter

The Kaplan–Meier curves were generated by Kaplan–Meier Plotter (URL: https://www.kmplot.com to analyze the prognostic value of EGR expression in BC (*Nagy et al., 2018*). The site contains the clinical information of 6,234 patients with BC. Each gene was divided into two groups: a high expression group and a low expression group, which were classified in accordance with median mRNA expression values. The required probe ids were then encoded into the database. Kaplan–Meier curves were subsequently plotted, and the overall survival (OS), recurrence-free survival (RFS), distant metastasis-free survival (DMFS), and post progression survival (PPS) of different genes in various BC subtypes were evaluated.

## Breast cancer gene–expression miner v4.2 (bcGenExMiner v4.2)

The Breast Cancer Gene-Expression Miner (bcGenExMiner) tool v4.2 (http://bcgenex.centregauducheau.fr/bc-gem/GEM-Accueil.php?js=1) is a statistical mining tool with published annotated transcriptome BC data (DNA microarray and RNA-seq data), which comprises 36 genomic data sets that are annotated (updated to January 2019) to analyze the relationship between EGR family mRNA expression levels and clinical parameters (age, ER, PR, HER2, etc.) (*Jézéquel et al., 2012*; *Jézéquel et al., 2013*).

## Statistical analysis

Using Student's $t$-test (*Hsu & Lee, 2010*) on the Oncomine data set for analysis, we established that differences with $p < 0.0001$ or 2.0-fold changes were statistically significant. BcGenExMiner v4.2 was used to conduct Tukey-Kramer's test, the Welch test and univariate Cox analysis.

# RESULTS

## The EGR mRNA expression levels in BC tissues

The mRNA expression levels of EGR family members in BC samples were compared with those in normal tissues (Fig. 1). The analysis showed that EGR1 and EGR2 were significantly downregulated in patients with BC in 17 (35.4%) and 14 (53.8%) datasets, respectively, and EGR3 was also significantly downregulated in four datasets. However, no relevant dataset indicated a difference in the expression level of EGR4. In the TCGA dataset, which contains 593 genetic samples from patients with BC, EGR1 expression was discovered in different invasive BCs and was significantly lower in BC (invasive ductal breast cancer: fold change $-14.944$; invasive lobular breast cancer: fold change $-7.001$; invasive breast cancer: fold change $-6.692$) (Table 1). According to the study by (*Curtis et al., 2012*), EGR1 was significantly downregulated in different pathological types of BC. In the TCGA dataset, EGR2 exhibited a fold change of $-3.173$ in invasive BC, while EGR3 showed a fold change of $-4.301$ in invasive BC.

## The EGR mRNA expression levels of patients with BC are significantly correlated with clinicopathological data

We used bcGenExMiner v4.2 to analyze the mRNA expression of the EGR family and various clinicopathological parameters and classified the data, as illustrated in Table 2. In ER-positive BC patients, EGR1/3 expression levels were significantly upregulated ($p < 0.0001$), while EGR2/4 expression levels were significantly downregulated (EGR2: $p = 0.0104$, EGR4: $p < 0.0001$). In PR-positive BC patients, EGR1/3 expression levels were significantly upregulated ($p < 0.0001$), and EGR4 was significantly downregulated (EGR4: $p = 0.0009$). Among BC patients with positive HER2, only EGR1 was significantly downregulated ($p = 0.0297$). When the nodule status was positive, EGR1/4 expression levels were significantly downregulated (EGR1: $p = 0.0338$, EGR4: $p = 0.0016$). However, when BC patients were over 51 years of age, there were no statistically significant differences in EGR1 mRNA expression, and EGR2/3/4 expression levels were significantly downregulated (EGR2: $p = 0.0387$, EGR3: $p = 0.002$; EGR4, $p = 0.0285$). The expression level of EGR1/4 in the triple-negative breast cancer (TNBC) group decreased ($p < 0.0001$).

## The different biological subtypes of BC and the influence of EGR expression levels on prognosis

Currently, the breast oncology field mainly classifies BC into four fixed biological subtypes: luminal A (ER+/HER2-/grade 1 or grade 2), luminal B (ER+/HER2-/grade 3), HER2 rich (any HER2+ tumor), and basal-like (ER-/PR-/HER2-). The treatment regimens and prognosis of the various biological subtypes are different (*Cejalvo et al., 2017*). Therefore,

| Analysis Type by Cancer | Cancer vs. Normal EGR1 | | Cancer vs. Normal EGR2 | | Cancer vs. Normal EGR3 | | Cancer vs. Normal EGR4 | |
|---|---|---|---|---|---|---|---|---|
| Bladder Cancer | | 3 | | 3 | | 1 | | |
| Brain and CNS Cancer | 1 | | 5 | | 1 | 4 | | 2 |
| Breast Cancer | | 17 | 1 | 14 | | 4 | | |
| Cervical Cancer | | 1 | | | | 1 | | |
| Colorectal Cancer | 1 | 1 | | | 2 | 1 | | |
| Esophageal Cancer | | | | | | | | |
| Gastric Cancer | | | 1 | | | | 1 | |
| Head and Neck Cancer | 1 | | 1 | 2 | 1 | | | |
| Kidney Cancer | | 2 | | | | | | |
| Leukemia | | 4 | | | | 1 | | |
| Liver Cancer | | 5 | | 2 | | | | |
| Lung Cancer | | 4 | | 3 | | 1 | 1 | |
| Lymphoma | | | 1 | | | 3 | | |
| Melanoma | | 2 | | | | | | |
| Myeloma | | | | | | 1 | | 1 |
| Other Cancer | 1 | 2 | 1 | | 1 | | | 5 |
| Ovarian Cancer | | 2 | | 1 | | 2 | | |
| Pancreatic Cancer | | | 1 | | | | | |
| Prostate Cancer | | | | 1 | 1 | | | |
| Sarcoma | | 5 | | | | | | |
| Significant Unique Analyses | 4 | 48 | 11 | 26 | 6 | 19 | 2 | 8 |
| Total Unique Analyses | 361 | | 345 | | 351 | | 310 | |

1 5 10    10 5 1
%

**Figure 1  The family of EGR mRNA expression levels in different cancer types.** Up: red, down: blue. $P < 0.05$, which confirms statistical significance. Color depth indicates the percentage of gene arrangement.

we analyzed the prognosis of the four underlying subtypes in relation to EGR expression using Kaplan–Meier Plotter. Analysis revealed that EGR1 was correlated with the three biological subtypes regarding RFS (Figs. 2B, 2D, 2E). The reduction in EGR1 was correlated with high OS and high RFS in the basal-like group (Figs. 2A, 2B) and was also related to high DMFS in patients who belonged to the HER2+ group(Fig. 2C). However, the high expression of EGR2 showed better RFS and DMFS rates in the luminal B group (Figs. 2F, 2H). In contrast, the low expression of EGR2 exhibited better PPS in the luminal B group (Fig. 2G). Highly expressed EGR3 showed good RFS in the basal-like group and the luminal B group(Figs. 2I, 2J). The above mentioned results are summarized in Fig. 2.

**Table 1**  **Datasets of EGR family in breast cancer.** Solie breast: *Sorlie et al. (2001)*; *Sorlie et al. (2003)*; Ma breast : *Saslow et al. (2007)* ; Curtis breast : *Curtis et al.(2012)*.

| Gene | Dataset | Type of BC vs. breast | Fold change | *P*-value | *t*-Test |
|------|---------|----------------------|-------------|-----------|----------|
| EGR1 | TCGA breast | Invasive BC | −6.692 | 1.35E–25 | −12.908 |
| | | Invasive Ductal BC | −14.944 | 3.23E–41 | −23.74 |
| | | Invasive Loublar BC | −7.001 | 5.60E–13 | −9.119 |
| | Solie breast | Ductal BC | −13.473 | 3.68E–20 | −20.081 |
| | Solie breast 2 | Loublar BC | −11.284 | 4.14E–05 | −10.364 |
| | | Ductal BC | −13.369 | 2.57E–09 | −20.244 |
| | Ma breast 4 | Ductal BC in situ epithelia | −7.073 | 6.62E–05 | −4.738 |
| | Richardson breast 2 | Ductal BC | −11.306 | 2.29E–12 | −12.684 |
| | Perou breast | Ductal BC | −14.193 | 1.37E–16 | −20.785 |
| | Curtis breast | Invasive Ductal BC | −6.866 | 2.02E–127 | −47.813 |
| | | Invasive Loublar BC | −4.156 | 2.54E–47 | −18.611 |
| | | Invasive BC | −5.165 | 8.43E–08 | −7.631 |
| | | Medullary BC | −9.539 | 6.13E–18 | −15.918 |
| | | Tubular BC | −4.04 | 1.45E–26 | −15.102 |
| | | Mucinous BC | −5.765 | 2.07E–18 | −13.128 |
| EGR2 | TCGA breast | Invasive BC | −3.172 | 3.34E–13 | −8.043 |
| | | Invasive Ductal BC | −5.759 | 3.17E–24 | −14.683 |
| | | Invasive Loublar BC | −3.723 | 1.43E–09 | −6.811 |
| | Curtis breast | Invasive Ductal BC | −5.022 | 9.45E–63 | −26.986 |
| | | Invasive Loublar BC | −3.803 | 2.41E–42 | −16.084 |
| | | Invasive BC | −4.413 | 5.08E–08 | −7.497 |
| | | Tubular BC | −3.917 | 1.40E–28 | −14.009 |
| | | Medullary BC | −4.301 | 6.68E–21 | −13.905 |
| | | Mucinous BC | −4.388 | 4.67E–15 | −10.103 |
| | Ma breast 4 | Ductal BC in situ stroma | −3.037 | 4.06E–07 | −6.737 |
| | | Ductal BC in situ epithelia | −15.213 | 1.32E–06 | −6.845 |
| | Richardson breast 2 | Ductal BC | −8.765 | 6.36E–09 | −12.409 |
| EGR3 | TCGA breast | Invasive BC | −4.301 | 1.26E–14 | −8.544 |
| | | Invasive Ductal BC | −8.619 | 8.09E–29 | −16.565 |
| | Curtis breast | Medullary BC | −2.249 | 3.92E–33 | −15.057 |
| | Richardson breast 2 | Ductal BC | −10.627 | 2.93E–06 | −7.733 |

According to the SBR classification criteria, the relationship between EGR expression level and BC progression was evaluated. The histological grading of breast cancer is closely related to prognosis. The Scarff-Bloom-Richardson (SBR) grading system, the most commonly used histological grading system at present, evaluates prognosis and guides chemotherapy by describing the differentiation degree of breast tumors (*Amat et al., 2002*; *Bansal et al., 2012*). The classification was determined according to three histological features of breast tumors: the glands and composition, the proportion of the nucleus of the gland pleomorphic, and the activity of nuclear fission. The overall rating of the final score together with the above three items were divided into three levels (I–III): grades 3∼5 were classified as level I, grades 6∼7 were classified as level II, and grades 8∼9 were classified as

Fei et al. (2019), *PeerJ*, DOI 10.7717/peerj.8183

**Table 2   Comparison of EGR expression betweenpatients with breast cancer and different clinicopathological parameters.** The number of patients included was determined by the EGR data set. ↑, up expression; ↓, lower expression.

| | EGR1 mRNA | | | EGR2 mRNA | | | EGR3 mRNA | | | EGR4 mRNA | | |
|---|---|---|---|---|---|---|---|---|---|---|---|---|
| | Numbers of | | | Numbers of | | | Numbers of | | | Numbers of | | |
| Variable | patients | Expression | *P*-value | patients | Expression | *P*-value | patients | Expression | *P*-value | patients | Expression | P-value |
| **ER,IHC** | | | | | | | | | | | | |
| + | 4,034 | ↑ | <.0001 | 3,632 | ↓ | 0.0104 | 3,855 | ↑ | <0.0001 | 3,391 | ↓ | <0.0001 |
| - | 1,586 | − | | 1,416 | − | | 1,529 | − | | 1,290 | − | |
| **PR,IHC** | | | | | | | | | | | | |
| + | 1,413 | ↑ | <0.0001 | 1,117 | − | 0.9,781 | 1,307 | ↑ | <0.0001 | 1,039 | ↑ | 0.0009 |
| - | 1,048 | − | | 776 | − | | 918 | − | | 718 | − | |
| **HER2,IHC** | | | | | | | | | | | | |
| + | 200 | ↓ | 0.0297 | 184 | − | 0.6,631 | 184 | − | 0.481 | 175 | − | 0.4,931 |
| - | 1,592 | − | | 1,405 | − | | 1,405 | − | | 1,365 | − | |
| **Nodal status** | | | | | | | | | | | | |
| + | 1,744 | ↓ | 0.0338 | 1,493 | − | 0.1,094 | 1,494 | − | 0.1,805 | 1,330 | ↓ | 0.0016 |
| - | 2,398 | − | | 2,399 | − | | 2,399 | − | | 2,232 | − | |
| **Age** | | | | | | | | | | | | |
| >51 | 2,212 | − | 0.431 | 2,093 | ↑ | 0.0387 | 2,094 | ↑ | 0.002 | 1,829 | ↑ | 0.0285 |
| >51 | 1,474 | − | | 1,343 | − | | 1,343 | − | | 1,235 | − | |
| **TriplE–negative status** | | | | | | | | | | | | |
| TNBC | 416 | ↓ | <0.0001 | 3,704 | − | 0.7,152 | 373 | ↓ | <0.0001 | 3,519 | ↓ | <0.0001 |
| Not TNBC | 4,133 | − | 373 | − | | 3,946 | − | | 361 | − | |

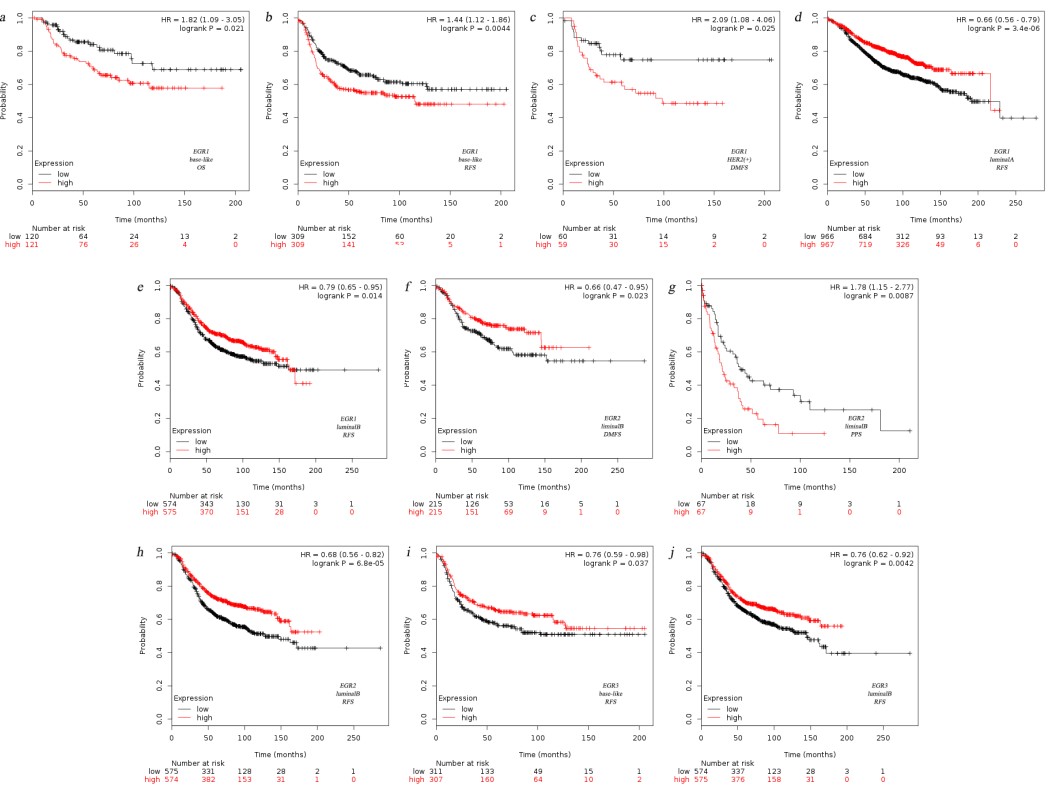

**Figure 2 Relationship between EGR family mRNA expression level and prognosis of BC patients.** OS, overall survival, RFS, completion free survival, DMFS, completion free survival, PPS, post progression survival.

level III. We used the bcGenExMiner v4.2 analysis tool to analyze the relation between the EGR expression levels and SBR grading of BC patients. Figure 3 illustrates the box plot results. According to the analysis results, all EGR expression levels are correlated with SBR classification. The decreased expression of EGR1, EGR2, and EGR3 indicated the progress of SBR grading. Additionally, the differences were statistically significant (Figs. 3A–3C: EGR1/3, $p < 0.0001$; EGR2, $p = 0.0007$). The increased expression of EGR4 suggested the progression of SBR grading (Fig. 3D: EGR4, $p < 0.0001$). The statistical results of the EGRs and SBR grading in each group were obtained using the Dunnett-Tukey-Kramer's test (Table 3). There was no significant difference between EGR2/4 expression and SBR classification.

## The association of EGR expression levels with survival

EGR1/2/3 mRNA expression levels were related to MRFS in BC patients with metastatic recurrence (Fig. 4). The prognostic analysis in bcGenExMiner v4.2 showed a correlation between EGR1/2/3 mRNA expression levels and metastatic MRFS in BC patients. Patients with increased EGR1/2/3 expression showed decreased MRFS (EGR1: HR 0.87; 95% CI [0.77–0.98], $p = 0.0212$; EGR2: HR, 0.81; 95% CI [0.71–0.92], $p = 0.0009$; EGR3: HR 0.78, 95% CI [0.69–0.89], $p = 0.0001$; Figs. 4A–4C). There was no correlation between the

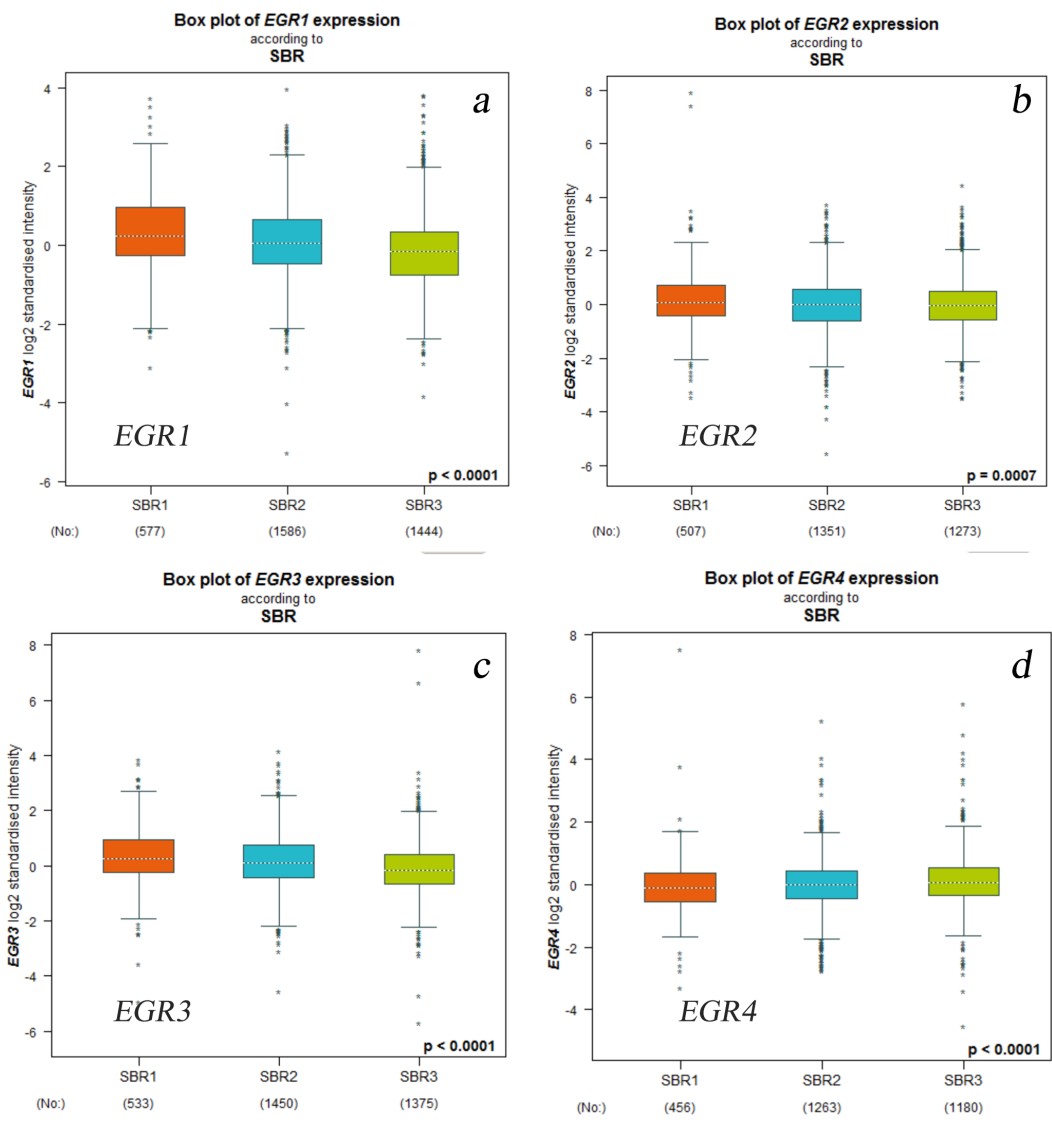

**Figure 3  Relationship between EGR family mRNA expression level and SBR gradingstatus.**
Welch's test was applied to generate $P$ value. Dunnett's Tukey's Kramer's test was used for pared-to-pared-comparison to evaluate the differences between groups. $P < 0.05$, which confirms statistical significance.(a) EGR1, (b) EGR2, (c) EGR3, (d) EGR4.

expression of EGR4 (HR 1.09; 95% CI [0.96–1.24], $p = 0.1660$) and MRFS in BC patients (Fig. 4D).

## EGR mutations in BC

We used the cBioPortal database for gene expression analysis and EGR family prognostic assessment. Among the 3,617 data samples, 165 (5%, data not shown) samples showed changes in EGR expression levels (0.6% of the samples showed changes in EGR1 expression, 1.4% of the samples showed changes in EGR2 expression, 2.9% of the samples showed changes in EGR3 expression, and 0.6% of the samples showed changes in EGR4 expression)

**Table 3  Dunnett–Tukey–Kramer's test for pairwise comparison in SBR criterion.** SBR, Scarff -Bloom - Richardson. *P* value, when $p < 0.05$, there is a significant statistical difference.

| mRNA | The comparison of SBR | The comparison of mRNA expression | *P*-value |
|------|------------------------|-----------------------------------|-----------|
| EGR1 | SBR1 vs. SBR2 | SBR2<SBR1 | <0.0001 |
|      | SBR2 vs. SBR3 | SBR3<SBR2 | <0.0001 |
|      | SBR1 vs. SBR3 | SBR3<SBR1 | <0.0001 |
| EGR2 | SBR1 vs. SBR2 | SBR2<SBR1 | <0.01 |
|      | SBR2 vs. SBR3 | SBR2=SBR3 | >0.10 |
|      | SBR1 vs. SBR3 | SBR3<SBR1 | <0.001 |
| EGR3 | SBR1 vs. SBR2 | SBR2<SBR1 | <0.01 |
|      | SBR2 vs. SBR3 | SBR3<SBR2 | <0.0001 |
|      | SBR1 vs. SBR3 | SBR3<SBR1 | <0.0001 |
| EGR4 | SBR1 vs. SBR2 | SBR1=SBR2 | >0.10 |
|      | SBR2 vs. SBR3 | SBR2<SBR3 | <0.01 |
|      | SBR1 vs. SBR3 | SBR1<SBR3 | <0.001 |

(Fig. 5A). In invasive BC patients, the expression of EGR1/2 was upregulated, while EGR3 expression was absent. Changes in EGR4 expression were nonsignificant. There was no obvious correlation with OS ($p = 0.813$, Fig. 5B). Moreover, there was no significant correlation between the number of copies of EGR3 and its mRNA expression level (Fig. 5C).

## DISCUSSION

EGRs belong to the early response group gene, which is a zinc finger transcription factor that can bind to GC-rich sequences in a limited manner (*Thiel & Cibelli, 2002*). EGRs possess a variety of biological functions. By regulating genes, EGRs enable cells to go through different stages of their life cycles and participate in cell proliferation, differentiation, apoptosis, and carcinogenesis according to specific cell types and under stimulation conditions (*Oliveira Fernandes & Tourtellotte, 2015*). EGR1 was first discovered in the screening of gene recognition upregulated by the addition of serum. The EGR2/3/4 genes were immediately discovered one year later (*Santino et al., 2017*). However, there have been no detailed reports on the association between the EGR family and the occurrence, development, and prognosis of BC. Therefore, this study carried out further evaluation and analysis.

EGR1 (one of the components of the early growth response family) can be activated instantaneously after being induced by various external stimuli (*Silverman & Collins, 1999*). As upstream and downstream molecules of various signaling pathways, EGR1 can regulate the expression of target genes. Shen et al. discovered that EGR1 can be stimulated and induced by various cytokines and hormones through the MAPK/ERK1/2 signaling pathway, thereby regulating the expression of target genes, which cause cell differentiation, apoptosis, and other pathophysiological processes (*Delmastro & Piganelli, 2011*)). Currently, the signaling pathway mediated by EGR1 is known to be of great significance in terms of the development of female reproductive organs (*Russell et al., 2003*). However, its specific role

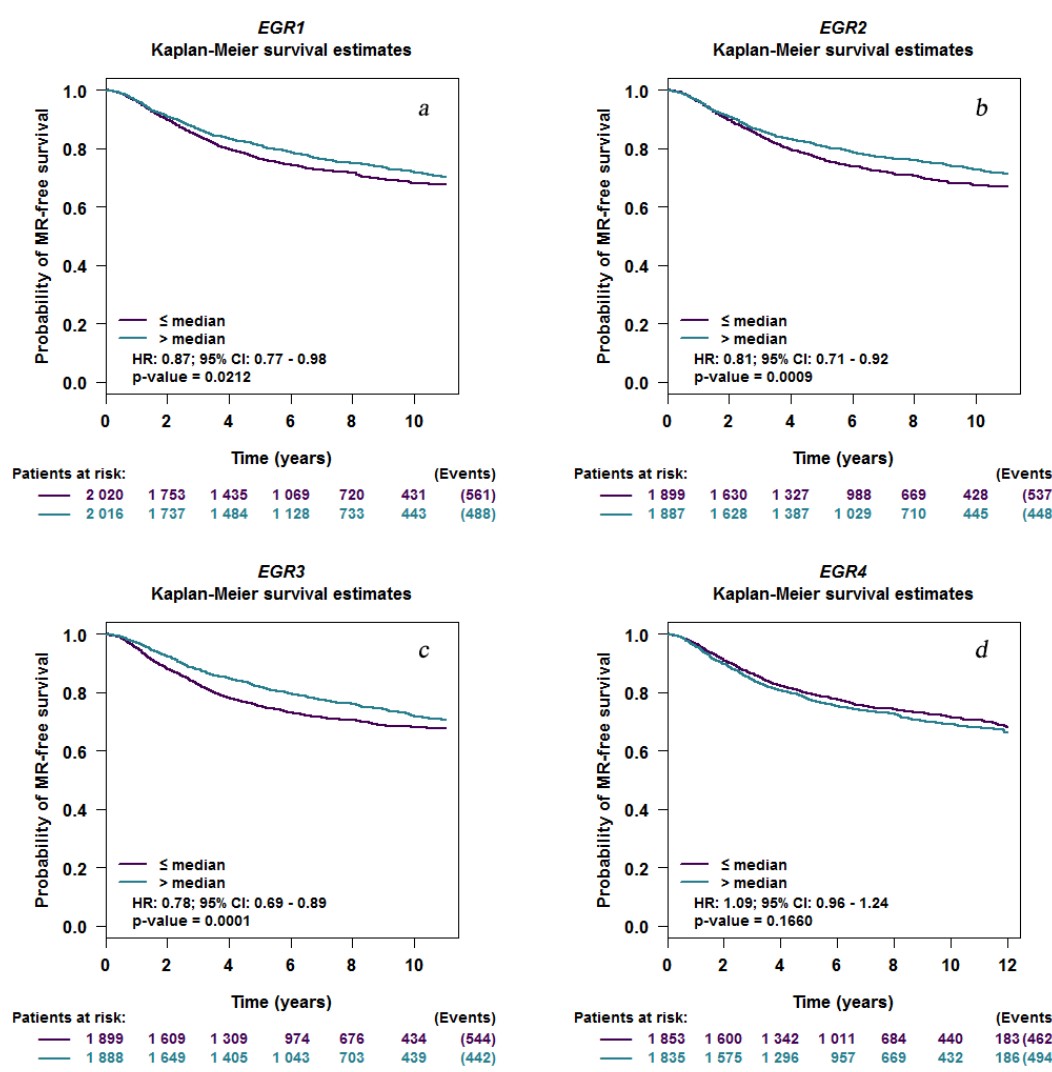

**Figure 4** **EGR mutations in patients with invasive breast cancer.** Kaplan–Meier curve depicting a positive association between the mRNA expression level of EGR1/2/3 and MR-free survival (EGR1: HR 0.87; 95% CI [0.77–0.98], $P = 0.0212$; EGR2: HR, 0.81; 95% CI [0.71–0.92], $P = 0.0009$; EGR3: HR 0.78, 95% CI [0.69–0.89], $P = 0.0001$). EGR4 expression level has no correlation with MR-free survival (HR 1.09; 95% CI [0.96–1.24], $P = 0.1660$). EGR, Early growth response proteins; HR, hazard ratio; CI, confidence interval; MR, metastatic relapse.(a) EGR1, (b) EGR2, (c) EGR3, (d) EGR4.

in BC is still unclear. In this study, the mRNA expression of EGR1 in BC samples was lower than that in normal breast tissues. Moreover, EGR1 expression was not consistent under different pathological characteristics: EGR1 was upregulated in ER(+) or PR(+) patients, while it was downregulated in HER2(−) patients. Further analysis on the classification of biological subtypes showed that EGR1 was correlated with the RFS of the four biological subtypes. The decrease in EGR1 expression indicates the progression of SBR classification; moreover, BC patients with higher EGR1 expression levels exhibited better OS and RFS. The above results provide some evidence for the follow-up treatment of BC patients.

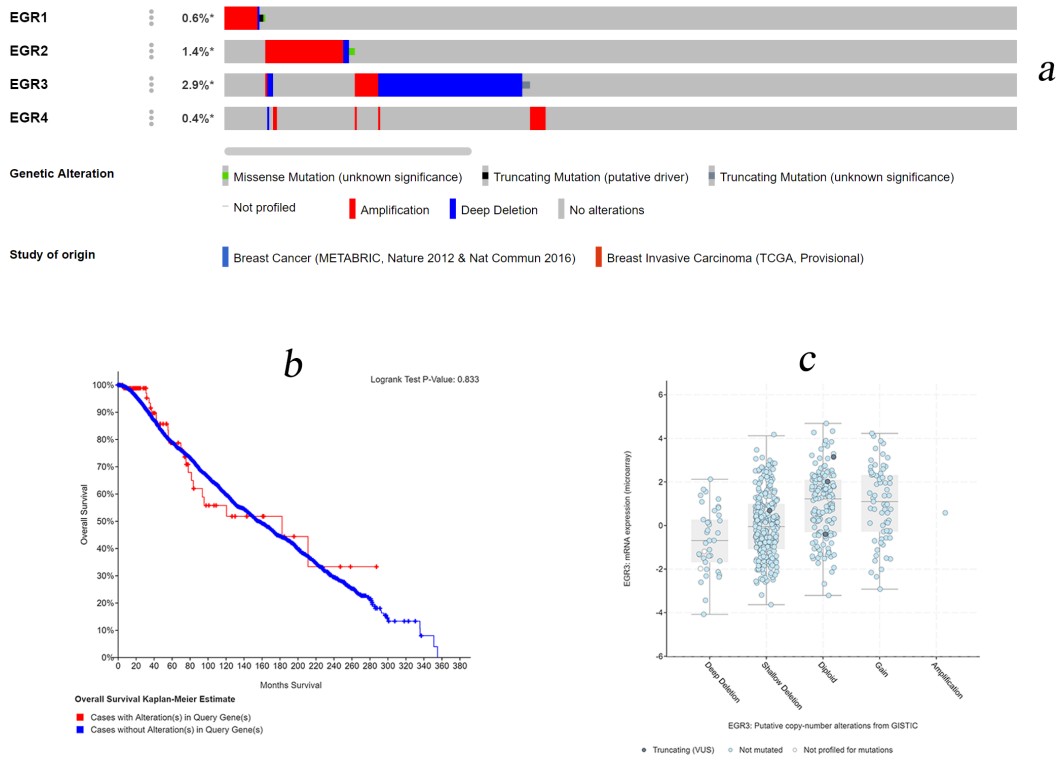

**Figure 5** **EGR mutations in patients with invasive breast cancer.** (A) Various genetic variations in the EGR family sample. Databases: Breast Cancer (METABRIC, Nature 2012 & Nat Commun 2016), Breast Invasive Carcinoma (TCGA, Provisional). (B) Overall survival rates with and without EGR3 copy-number alternations. Logrank Test *P*-Value: 0.833. (C) The relationship between the number of copies of EGR3 and its mRNA expression level.

The role of EGR2 in the central/peripheral nervous system has been widely reported (*LeBlanc, 2005*; *Nonchev et al., 1996*). In addition, EGR2 can negatively regulate the activation of T-cells and B-cells as well as the production of proinflammatory cytokines by inducing and inhibiting the expression of some cytokine signaling molecules (*Li et al., 2013*). Earlier reports have verified the involvement of EGR2 in the occurrence, invasion, and migration of a wide variety of tumors. Chen et al. found that the regulation of EGR2 expression by the competitive combination of LINC01939 with mir-17-5p may inhibit the metastasis and EMT of gastric cancer (*Chen et al., 2019*). EGR2 can reduce the phosphorylation of JAK2 and STAT3 by regulating the expression of SOCS-1 (*Lu et al., 2017*). In addition, a lack of EGR2 results in defective cloning amplification of T-cells as a response to viral infection, with overactivation and overdifferentiation (*Miao et al., 2017*). The regulatory effect of EGR2 on T-cells is crucial for maintaining immune homeostasis. Additionally, the regulation of EGR2 expression can control the immune regulatory pathway and avoid the occurrence of tumor immune escape. In the current study, it was discovered that EGR2 expression was significantly decreased in BC patients (including patients with recurrence and metastasis), and a low EGR2 expression level resulted in poor MRFS. This study found that the mRNA expression level of EGR2 in the samples of BC

patients was lower than that in normal breast tissues. In addition, the upregulation of EGR2 mRNA expression in BC patients resulted in better OS and RFS, which provides a basis for EGR2 to become a biological marker for evaluating the prognosis of BC patients.

EGR3 has been reported in various cancers. As a direct target of mir-71, EGR3 is negatively regulated and downregulated to promote the migration and invasion of HCC cells (*Wang et al., 2017*). Other studies have discovered that the expression level of EGR3 mRNA in prostate cancer samples is high, which can be used as a marker for cancer diagnosis and as a prognostic indicator that distinguishes between invasive and noninvasive tumors (*Rebecca et al., 2013*). EGR3 activates related inflammatory signaling pathways (such as the NF-kb pathway) by activating the expression of IL-6 and IL-8, which are closely related to the occurrence and development of cancer (*Baron et al., 2014*). EGR3 has been shown to play an important role in the invasion of BC (*Suzuki et al., 2007*). According to the analysis of this study, the expression of EGR3 mRNA was downregulated in BC and even absent in BC patients with metastasis and recurrence. Moreover, EGR3 was correlated with RFS in BC patients. We also found that the higher the SBR grade is, the lower the expression level of EGR3 mRNA. In addition, EGR3 mRNA expression was significantly correlated with BC MRFS. Moreover, it was an independent prognostic factor for BC. Therefore, EGR3 can be utilized as a biological marker for BC diagnosis and is an important indicator for prognosis.

There are few reports on EGR4 in tumors. At present, some literature suggests that EGR4 may be an oncogene that promotes the development of NSCLC (*He et al., 2019*; *Matsuo et al., 2014*). However, the relationship between EGR4 and the occurrence, development, and prognosis of BC has not been reported. In this study, the upregulated mRNA expression level of EGR4 was correlated with SBR grading but not significantly correlated with OS and RFS in BC patients. Therefore, EGR4 may be a potential oncogene in BC.

This analysis helped further our understanding of the expression level and prognostic value of the EGR family in BC and provided some evidence for the family members as new prognostic biomarkers or promising therapeutic targets for BC. However, we have focused on only the mRNA expression level and prognostic value of this family, without further analysis of its protein expression level and some related signaling pathways. Additional studies will explore the potential molecular mechanism of EGR in BC.

The limitations of this study should be noted. First, screening for biomarkers was based on statistical methods rather than biological experiments. Second, further conclusions need to be carried out in vitro and further validated in prospective studies and multicenter clinical trials. Third, due to the different emphases of different databases and different included studies, this study cannot guarantee the comprehensive application of data from different data sources and different databases, so the sensitivity and specificity of the data analysis results will also be different.

Generally, the EGRs might be involved in the occurrence and development of BC. At present, there are very few studies on the EGR family in BC, and they are not systematic. No expression differences of the EGR family members in BC in the literature were discussed, and the specific mechanism is unclear.

## CONCLUSIONS

This study specifically studied the expression of EGRs in BC and evaluated its clinical and prognostic value. The data analysis results suggest that EGR2/3 may be a potential diagnostic marker for BC, which can provide a basis for the prognostic assessment of BC. EGR4 may play a role in the occurrence and development of BC. Nevertheless, our research still has shortcomings. Currently, it is limited to database mining, and further in vitro experiments will be conducted based on the above conclusions in the future.

## ACKNOWLEDGEMENTS

We express our appreciation to Oncomine (our data contributor), Kaplan–Meier Plotter, and bcGenExMiner v4.2, cBioPortal.

### Funding

The present study in funded by the Zhejiang Provincial Natural Science Foundation of China (grant no. LY20H270009)the Zhejiang Provincial Project for the key discipline of traditional Chinese Medicine (Yong GUO, no.2017-XK-A09), the Zhejiang Province Famous Old Chinese Medicine Academic Inheritance and Specialty Construction project; Guo Yong of Zhejiang Province Famous Traditional Chinese Medicine Academic Experience Inheritance and Specialist Construction; and the Zhejiang University of Traditional Chinese Medicine Youth Scientific Research Innovation Fund, (grant no. KC201929). The funders had no role in study design, data collection and analysis, decision to publish, or preparation of the manuscript.

### Grant Disclosures

The following grant information was disclosed by the authors:
Zhejiang Provincial Natural Science Foundation of China: LY20H270009.
Zhejiang Provincial Project for the key discipline of traditional Chinese Medicine (Yong GUO): 2017-XK-A09.
Zhejiang Province Famous Old Chinese Medicine Academic Inheritance and Specialty Construction project; Guo Yong of Zhejiang Province Famous Traditional Chinese Medicine Academic Experience Inheritance and Specialist Construction.
Zhejiang University of Traditional Chinese Medicine Youth Scientific Research Innovation Fund: KC201929.

### Competing Interests

The authors declare there are no competing interests.

### Author Contributions

- Yuchang Fei conceived and designed the experiments, performed the experiments, analyzed the data, prepared figures and/or tables, approved the final draft.
- Huan Yu analyzed the data, prepared figures and/or tables, approved the final draft.

- Shuo Huang performed the experiments, prepared figures and/or tables, approved the final draft.
- Peifeng Chen conceived and designed the experiments, contributed reagents/materials/analysis tools, authored or reviewed drafts of the paper, approved the final draft.
- Lei Pan analyzed the data, contributed reagents/materials/analysis tools, authored or reviewed drafts of the paper, approved the final draft.

## Data Availability

This study used publicly-available data which was downloaded from https://www.oncomine.org/resource/login.html (Dataset: TCGA breast, Solie breast, Ma breast, Richardson breast, Perou breast, Curtis breast).

https://www.cbioportal.org/ (Dataset: Breast Cancer (METABRIC, Nature 2012 & Nat Commun 2016, Breast Invasive Carcinoma (TCGA, Provisional)).

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
