# Peer review of "Expression and prognostic analyses of early growth response proteins (EGRs) in human breast carcinoma based on database analysis"

_PeerJ, doi:10.7717/peerj.8183_

## Round 0.1 · original submission · Major Revisions

Your manuscript has been reviewed and requires modifications prior to making a decision.

The comments of the reviewer(s) are included at the bottom of this letter.
Reviewers indicated that the methods section and interpretation of the results

should be improved. I agree with this evaluation and I would, therefore, request for the manuscript to be revised accordingly.

I would also like to suggest the following changes:

-The manuscript also needs extensive English editing since there are several typos and grammatical errors.

-It is not clear how the authors checked normality assumption of the data? Provide the test name.

Reviewer 1 ·

Basic reporting

By using publicly available datasets, Fei et al. analyzed the mRNA expression of the Early Growth Factors (EGR) 1 to 4 in breast cancer samples and their correlation with survival in those patients.

• Professional English and proper punctuation should be used through out the text.
• The acronym “BC” for Breast Cancer should be spell it out the first time the term is used in the text (abstract).
• Briefly clarify in the text the meaning of SBR grading 1-3 and include a reference that includes more details about it.
• Parts of the discussion section are introducing the EGRs, which should be in the introduction section.
• Figures 4 and 5 are poorly explained in the results and the legends do not describe the figures. Overall proper figure legends should be added.
• Lines 37, 51: please add referring literature supporting the data provided.
• Include citation of Oncomine: Rhodes DR, Yu J, Shanker K, Deshpande N, Varambally R, Ghosh D, Barrette T, Pandey A, Chinnaiyan AM. 2004. Oncomine: a cancer microarray database and integrated data-mining platform. Neoplasia 6:1–6
• Include citation of kmplot: Nagy A, Lánczky A, Menyhárt O, Győrffy B. Validation of miRNA prognostic power in hepatocellular carcinoma using expression data of independent datasets, Scientific Reports, 2018;8:9227

Experimental design

• Oncomine data needs to be re-analyzed to contain the most up-to-date datasets or add the date this analysis was last performed (I performed an Oncomine analysis with EGR1-4, which showed different results compared to the one presented on Fig. 1.)
• Threshold Fold Change used for Oncomine analysis should be included in text (line 72)
• Line 106, please add ratio or percentage of studies. For example: EGR1 was downregulated in 17 out of 48 Unique Analysis (~35%)
• Include more details on how SBR grading and EGR expression was calculated.
• Data from figures 4 and 5 are not explained in the text.

Validity of the findings

There seems to be a correlation between low EGRs in breast cancer tumors compared to normal tissues. However, based on this metanalyses, EGR mRNA levels does not seem to be consistent to be used for prognostics as suggested by the authors.

Reviewer 2 ·

Basic reporting

* Abstract: "the survival data of breast cancer patients were compared to the transcriptional expression levels of the four proteins": The authors mean transcription exp levels were compared in BC patients ...... the structure of the sentence is wrong and can be rephrased to "the association between the survival data and tc levels were investigated". Also, the mentioned four proteins were not identified.

* Abstract: "In addition, The mRNA expression level of EGR1/2/3 were related to MRFS in BC patients with metastatic recurrence". What does M in MRFS mean and "related": What is the direction of the p value?

* Abstract: "EGR4 may have played a role in the development of BC". However, in the results section, the authors mentioned "However, EGR4 has no relevant dataset to indicate a difference in expression level". This is confusing.

* Introduction: The authors reviewed some literature of EGR proteins; however, they did not clarify the gaps in the literature and build the rationale for their study.

* Discussion: Although this section is more organized than the results section, it did not discuss in depth the authors' results and talked a lot about the classification grade and different types of survival. Also, they mentioned up and down regulation in many subtypes without adequate discussion.

Experimental design

- The authors should further describe the TCGA database: how they retrieve BC samples, how many studies they retrieve, any special filtration parameters they used, all alive or including dead, certain race, certain histopathology, certain types of studies, certain gender, etc, they removed patients with unknown or missing data? ... BC has the largest no of samples in tcga and hence has a lot of variations.

- Define the abbreviation the first time they are mentioned: OS, RFS, DMFS, PPS

- Line 89: Put a link for bcGenExMiner v4.2

- Line 97: "or 1.5-fold changes of the difference was statistically significant" Reference?

- Lines 102 and 103 of the results section should be moved to the methods section. Same for lines 128-133.

Validity of the findings

- The results section: should be divided into sections e.g (Expression level in BC tissues, then Association with pathological features, then association with survival) to be more clear.

- You should organize your results by expression levels, not by database. This will enable you to compare expression levels for the same receptor from different databases.

- I did not see classifications by grade except in pathological grade (intermediate and poor differentiation grade) and this could be classified within all 4 molecular subtypes mentioned: luminal A, B, TNBC, and Her2. Also, the results structure is confusing and should be divided and structured better.

- Line 108: In the TCGA dataset which contains 593 EGR1 genetic samples of patients with BC": remove EGR1 from this sentence.

- Lines 146 and 147: "to analyze the EGRs and BC of SBR" should be "to analyze the relation between EGR expression levels and SBR grading of BC patients"

- Line 147: "box-type graph results": It is called "box plot"

- Line 154: This should be a new section with a new title "EGR mutation in BC"

- Table 1: Solie breast, Ma breast, curtis breast: Add in the footer the explanation of these names of reference.

Additional comments

- Although the overall language is good, the manuscript needs revision by an expert language editing service. Also, lack or excess of spacing sometimes should be fixed (line 69, 79, and 239).

- Some sections in the manuscript need better organization, especially the results section.

- The figures are good illustrations, but some has low resolution.

Reviewer 3 ·

Basic reporting

English style needs to be improved. Some sentences need to be rephrased.
References are fine, background is ok.
Article is soundly structured.

Experimental design

Research questions need to be further investigated.
Specific comments:
1) Metabric database needs to be included in the analyses.
2) EGR transcription factors expression needs to be stratified following the main subtypes of BC.
3) Since EGR are transcription factors with well known target genes what about the expression of EGR gene signatures in BC subtypes

Validity of the findings

Findings are valid but insufficient to support authors conclusions

Additional comments

The manuscript in its present form is not suitable for publication, it needs further experimental work as previously mentioned.

Reviewer 4 ·

Basic reporting

In this manuscript, the authors combine the data from previous studies on early growth response proteins (EGRs) in breast cancer, by taking advantage of the analysis tools incorporated in the selected databases and draw their conclusions.

- Unfortunately, the manuscript is written in poor English and needs extensive proofreading before it may be published as a review.

- Additionally, the manuscript lacks the professional article structure of the subheadings. Very crowded data presented as a single paragraph with an extensive number of abbreviations which the authors did not spell out at their first mention in the text or spelled them incorrectly (e.g. "RFS: completion free survival, DMFS: completion free survival" among others).
All these add difficulty to follow the authors in their elaborations

Experimental design

- Overall, the study does neither report any new findings (not provided to be novel) nor independently validate any of the previously published findings by wet-lab results.

- The authors should pay attention that the applied databases have different depth of coverage and pipelines for data calling with feature different sensitivity and specificity.

- Overall the paper qualifies as a review rather than an original research article.

Validity of the findings

- Not tested in the current work as mentioned above. The work is lacking a validation step in their lab to add to the previous knowledge.

Additional comments

In addition to the above comments:
- By the end of the introduction, the authors stated in their aim that “Additionally, its expression in breast cancer was analyzed to determine its value in the treatment and diagnosis of breast cancer”. There was no evaluating analysis for diagnostic value assessment (e.g. ROC curve analysis) or the treatment response all over the manuscript.

- The title of the manuscript is not precisely describing the real work by the authors.

---

## Round 0.2 · accepted · Accept

Thank you very much for the submission of a revised version of your paper. I have gone through the revised, track-changes manuscript and rebuttal letter, and see that the authors addressed the reviewers' concerns and substantially improved the content of the manuscript. So, based on my own assessment as an editor, no further revisions are required, and the manuscript may be now accepted for publication in its current form.

Reviewer 1 ·

Basic reporting

English was clearly improved and proper literature was included. Tables and graphs were modified to be clear. The tracked changes manuscript does not contain all the modifications made, only the major additions.

Experimental design

The authors covered most of the reviewers points.

Validity of the findings

The authors explicitly included in title and text this work limitations (metanalyses only without experimental data).

Reviewer 2 ·

Basic reporting

The authors have adequately addressed my concerns in this section.

Experimental design

The authors have adequately addressed my concerns in this section.

Validity of the findings

The authors have adequately addressed my concerns in this section.

Additional comments

The authors have adequately addressed my concerns in this section.

Reviewer 4 ·

Basic reporting

- Clear with great improvement in the language.
- The text was enriched with the appropriate citations.
- The structure has been improved all over the manuscript

Experimental design

- It is now clear how this work fills an identified knowledge gap with some limitations that have been mentioned by the end of the manuscript.

- The description of the methods has been improved with sufficient details and information to replicate.

Validity of the findings

- Detailed work has been provided.
- Conclusions are well stated and linked to the original research question.

Additional comments

- The authors have adequately addressed the concerns raised by the reviewer. Thank you